# Squalene in oil-based adjuvant improves the immunogenicity of SARS-CoV-2 RBD and confirms safety in animal models

Ricardo Choque-Guevara[1], Astrid Poma-Acevedo[1], Ricardo Montesinos-Millán[1], Dora Rios-Matos[1], Kristel Gutiérrez-Manchay[1], Angela Montalvan-Avalos[1], Stefany Quiñones-Garcia[1,2], Maria de Grecia Cauti-Mendoza[1,2], Andres Agurto-Arteaga[1], Ingrid Ramirez-Ortiz[1], Manuel Criollo-Orozco[1], Edison Huaccachi-Gonzales[1], Yomara K. Romero[2], Norma Perez-Martinez[1], Gisela Isasi-Rivas[1], Yacory Sernaque-Aguilar[1], Doris Villanueva-Pérez[1], Freddy Ygnacio[1], Katherine Vallejos-Sánchez[2], Manolo Fernández-Sánchez[1], Luis A. Guevara-Sarmiento[1], Manolo Fernández-Díaz[1], Mirko Zimic[1,2]*, for the COVID-19 Working Group in Perú[¶]

**1** Laboratorios de investigación y desarrollo, FARVET SAC, Chincha, Ica, Perú, **2** Laboratorio de Bioinformática, Biología Molecular y Desarrollos Tecnológicos, Laboratorios de Investigación y Desarrollo, Facultad de Ciencias y Filosofía, Universidad Peruana Cayetano Heredia, Lima, Peru

¶ Membership of the COVID-19 Working Group in Peru is provided in the Acknowledgments.
* mirko.zimic@upch.pe

**Data Availability Statement:** All relevant data are within the article and its Supporting information files.

## Abstract

COVID-19 pandemic has accelerated the development of vaccines against its etiologic agent, SARS-CoV-2. However, the emergence of new variants of the virus lead to the generation of new alternatives to improve the current sub-unit vaccines in development. In the present report, the immunogenicity of the Spike RBD of SARS-CoV-2 formulated with an oil-in-water emulsion and a water-in-oil emulsion with squalene was evaluated in mice and hamsters. The RBD protein was expressed in insect cells and purified by chromatography until >95% purity. The protein was shown to have the appropriate folding as determined by ELISA and flow cytometry binding assays to its receptor, as well as by its detection by hamster immune anti-S1 sera under non-reducing conditions. In immunization assays, although the cellular immune response elicited by both adjuvants were similar, the formulation based in water-in-oil emulsion and squalene generated an earlier humoral response as determined by ELISA. Similarly, this formulation was able to stimulate neutralizing antibodies in hamsters. The vaccine candidate was shown to be safe, as demonstrated by the histopathological analysis in lungs, liver and kidney. These results have shown the potential of this formulation vaccine to be evaluated in a challenge against SARS-CoV-2 and determine its ability to confer protection.

## Introduction

In December 2019, a phylogenetically related SARS-CoV virus, later identified as SARS-CoV-2, caused an outbreak of atypical pneumonia in Wuhan. This virus is associated with a high rate of transmission, the appearance of symptoms such as fever and respiratory difficulties

**Funding:** This study was funded/ supported by Laboratorios de Investigación y Desarrollo - FARVET and partially by Fondo Nacional de Desarrollo Científico, Tecnológico y de Innovación Tecnológica - FONDECYT (https://www.fondecyt. gob.pe/) under the contract 060-2020-FONDECYT. MFD and MZ were granted by Consejo Nacional de Ciencia, Tecnología e Innovación Tecnológica (CONCYTEC). These funder supported salaries for RCG, RMM, APA, DRM, KGM, AM, SQG, MCM, AAAM, IRO, MCO, EHG, NPM, GIR, YSA and DVP and supplied materials for the study. The funders had no role in study desing, data collection and analysis, decision to publish, or preparation of manuscript.

**Competing interests:** The authors have declared that no competing interests exist.

leads later to pulmonary and systemic failure with an exacerbated inflammatory condition that can lead to death [1]. The high transmission and mortality, coupled with the lack of effective treatment, justify the urgent for development of vaccine candidates.

SARS-CoV-2 recognizes the Angiotensin Converting Enzyme-2 (ACE-2), which belongs to the surface of several types of human cells. The glycosylated Spike (S) protein gives the virus the ability to bind to the cell membrane and then fuse for the entry of viral RNA. The Spike protein has the S1 domain, and at its most distal end has a receptor binding sub-domain (RBD) [2]. The RBD is responsible for the binding of the virus to the ACE-2 receptor of host cells [3, 4]. The amino acid sequences of RBD protein are being subjected to a positive selective pressure, which is conferring greater affinity to the receptor, this is due to the change in the structural conformation of the ACE-2 binding motif [5]. An important mechanism of neutralization is the blockade of ACE-2 binding to the virus, so candidate vaccines based on the RBD domain induce a strong immune response, generating a remarkable humoral and cellular immunity [6–8].

Several vaccine candidates use the baculovirus expression system. This system is used widely due to its easy manipulation and the ability to produce complex proteins with suitable glycosylation patterns [9]. Currently, several human and veterinary vaccines manufactured in this system are commercialized [10] and produced in large-scale for clinical trials [11, 12]. However, these vaccines require an appropriate adjuvant to stimulate a strong immune response.

There are several types of adjuvants on the market, which have an immunogenic effect when inoculated in animals and humans: those that are based on Alum [13], as well as emulsions based on mineral or non-mineral oils [14], which are the most widely used and approved for use in humans [15]. Alum-based adjuvants are not highly effective in stimulating the cellular immune response of either Th1 or Th2 [16]. These adjuvants require improvements in their concentration and the type of aluminum used to generate a cellular-type immune response; however, these could cause necrosis or tissue damage in the inoculation area [17]. This has led to the use of emulsions based on squalene-in-water, which come in formulations according to the interface where they are prepared: oil-in-water (O/W), which are microdroplets of oil in the aqueous phase together with the antigen; and water-in-oil (W/O), microdroplets of water containing the antigen, in an oily phase [18].

In the present study, a commercial O/W adjuvant and a proprietary W/O adjuvant were mixed with a purified RBD and administered through intramuscular route to evaluate its immunogenicity and safety in mice and hamsters.

## Materials and methods

### Animals

This study used thirty-five female albino mice (*Mus musculus*) strain BALB/c of 5–8 weeks-old and 5 female Golden Syrian hamsters (*Mesocricetus auratus*) of 8–10 weeks-old obtained from the Universidad peruana Cayetano Heredia (UPCH) and the Instituto Nacional de Salud (INS-Perú), respectively. This study was carried out in strict accordance with the recommendations described for use and animals care of the INS-Perú [19].

### Adjuvants

An oil-in-water (ESSAI 1849101) hereinafter defined as A1, and a modified adjuvant resulting from a mix of water-in-oil adjuvant and squalene (Industrial secret—FARVET company) hereinafter defined as A3 were used.

## Ethics statement

The use of animals was aligned to ethical protocols approved by the Bioethics Committee of the Universidad Nacional Hermilio Valdizán and the animal's ethical Committee at the Universidad Peruana Cayetano Heredia, registered as approval certificates of Research Project No. 1, 2, and 10 and E011-06-20, respectively. Animal immunizations and procedures were performed by qualified personnel following the ARRIVE guidelines [20]. The animals were euthanized by trained veterinary personnel following the guideline stablished by the American Veterinary Medical Association (AVMA) [21]. Briefly, mice were euthanized by anesthetic overdose, inoculating 200 μL of a ketamine (100 mg/mL), xylazine (20 mg/mL) and atropine (1 mg/mL) solution using a hypodermic needle by intramuscular route. The procedure was performed rapid in order to minimize the suffering. The animal was kept in a quiet place until the effects of anesthesia began to manifest.

## Recombinant RBD expression in Sf9 cells

**Recombinant baculovirus generation.**    The amino acid sequence of the SARS-CoV-2 spike protein was obtained from the SARS-CoV-2 reference genome Wuhan-Hu-1 (Genbank accession number: NC_045512.2). For the design of RBD construct, the Pro330-Ser530 region was selected. The sequence was optimized for expression in insect cells, the gp67 secretion signal peptide was added at the N-terminal and a 6xHis-tag in the C-terminal region. The resulting sequence was chemically synthesized by GenScript Laboratories and cloned at the EcoRI/HindIII sites of pFastBac1 (Thermo Fisher Scientific, USA) under the control of the polyhedrin promoter and upstream of the SV40 polyadenylation sequence. Transformation of competent DH10BAC cells and transfection of Sf9 cells were performed with the Bac-to-Bac technology following the manufacturer's instructions (Thermo Fisher Scientific, USA).

**Propagation of baculovirus and expression of RBD in Sf9 insect cells culture.**    The recombinant baculovirus was amplified in Sf9 cells (Thermo Fisher Scientific, USA) to a density of $2 \times 10^6$ cells/mL in ExCell 420 medium (Sigma Aldrich, USA) supplemented with 5% fetal bovine serum (Gibco, USA). Cultures were infected at a multiplicity of infection (MOI) of 0.4. At 48 hours post infection (hpi), cultures were centrifuged at 4500 rpm for 15 minutes. The supernatants were collected and titrated by plaque assay. Viral stocks were stored at 4°C until use.

For protein production, 7 L of Sf9 cell culture at a density of $2 \times 10^6$ cells/mL were infected with the baculovirus at a MOI of 3 using a Biostat B plus bioreactor (Sartorius, Germany). The following conditions were maintained during the culture period: temperature at 28°C, pH at 6.2, 50% dissolved oxygen (DO) with an oxygen flow rate of 0.1 vvm via micro sparger and agitation at 150 rpm. At 48 hours post-infection, the cultures were centrifuged at 4500 rpm for 15 minutes and the supernatant was filtered through a 0.22 μm membrane.

## Recombinant RBD purification

**Tangential filtering.**    Tangential filtration was conducted on a Hydrosart cassette (Sartorius, Germany) with 5 kDa of nominal molecular weight cutoff (MWCO) on a SARTOFLOW Advanced (Sartorius, Germany) tangential flow system. The supernatant was retained and concentrated to a volume of 2 L. Subsequently, the retentate was diafiltered into a saline phosphate buffer (PBS) at pH 6.3 and concentrated again to a volume of 1 L, filtered through 0.22 μM membrane and stored at 4°C until use.

**Affinity chromatography.**    As a first step, an immobilized metal affinity chromatography (IMAC) was performed using a HisTrap Excel column (1.6 x 2.5 cm) on an AKTA Pure 25 L system (Cytiva, Sweden). Desalting and buffer exchange were performed on a Hiprep 26/10

desalting column (Cytiva, Sweden) using PBS pH 7.4 throughout the elution phase. The desalted protein was concentrated on an Amicon 10,000 MWCO (Merck, Germany) and filtered through a 0.22 μM membrane.

**Size exclusion chromatography.**   As a second step, a size exclusion chromatography was performed on a Superdex 200 increase 10/300 GL column (Cytiva, Sweden) using PBS pH 7.4 during the entire process. Protein fractions were collected and analyzed by SDS-PAGE under reducing conditions and Western blot using a commercial anti-His monoclonal antibody. The pool of selected fractions was concentrated using an Amicon 10,000 MWCO (Merck, Germany) and filtered through a 0.22 μM membrane. The concentration of purified RBD was determined using the Bradford assay (Merck, Germany), following the manufacturer's instructions.

## Recombinant RBD characterization in vitro

**RBD binding to human ACE-2.**   A 96-well plate was coated overnight at 4ºC with 100 μL of a recombinant human ACE-2 fused to a Fc fragment (GenScript Laboratories, USA) at 1 μg/mL in carbonate buffer (pH 9.6). The plate was blocked with 3% skimmed milk for 1 hour at room temperature (RT) and then washed five times with PBS 0.05% Tween 20 (PBS-T). Serial dilutions (1:2) of purified RBD were performed in PBS, starting from 2 μg/mL and ending to 1.9 ng/mL. Dilutions were added to the wells and incubated for 2 hours at 37ºC. Five washing steps with PBS-T were performed and then, 100 μL of rabbit IgG polyclonal anti-spike antibody (SinoBiological, China) was added to the wells (1:5000) in 1% skimmed milk and incubated for 1 hour at 37ºC. The plate was washed five times with PBS-T. Then, 100 μL of anti-rabbit IgG HRP conjugated (GenScript Laboratories, USA) (1:30,000) in 1% skimmed milk was added to the wells. The plates were incubated at 37ºC for 1 hour. Finally, the plates were washed with PBS-T five times, and 100 μL of TMB (Sigma Aldrich, USA) were added to the wells and incubated for 15 minutes at RT. The reaction was stopped with 50 μL of 2N sulfuric acid and the absorbance at 450 nm was read with an Epoch 2 microplate reader (Biotek, USA).

**RBD binding to Vero-E6 cells.**   Vero-E6 cells (Cod. CRL-1586™, ATCC®, USA), which were previously cultured in DMEM/F12 (HyClone, USA) + 10% fetal bovine serum (FBS) (HyClone, USA), were harvested and washed with DPBS with 5% FBS (FACS buffer). Approximately $10^6$ cells were blocked with FACS buffer and 5% of normal mouse serum (Abcam, USA) for 30 min at 37˚C. Then, the cells were incubated with the purified RBD (8 μg/mL) for 2 h at 37˚C. To remove the excess of RBD not attached to Vero E6, the cells were washed with FACS buffer twice. After that, the mix was marked with rabbit monoclonal antibody anti-SARS-CoV-2 S1 (1:200) (Sino Biological, China) as the primary antibody for 1 h at 37˚C, followed by the addition of the secondary goat anti-rabbit IgG antibody conjugated with Alexa Fluor 488 (1:200) (Abcam, USA). Finally, cells were acquired by the BD FACSCanto™ II flow cytometer (BD Biosciences, USA). The data was analyzed using the software FlowJo v.10.6 (BD Biosciences, USA), and the graphics were generated with GraphPad Prism 8.0.1. For the interpretation of results, the percentage of positive cells indicates the binding of RBD to Vero E6 cells.

**RBD recognition by immunized sera.**   Purified RBD was loaded at 0.2 μg/well and electrophoretically separated by SDS-PAGE under non-reducing conditions and transferred to nitrocellulose membranes using an e-blot device (GenScript Laboratories, USA). The membranes were blocked with 5% (w/v) non-fat milk in PBS with 0.1% of Tween 20 at pH 7.4 and incubated overnight at RT. Then, membranes were washed three times for 5 minutes each with Tris-buffered saline containing 0.1% (v/v) Tween 20 (TBS-T) and incubated for two

hours at RT with serum of a hamster immunized with a Newcastle disease virus expressing the S1 sub-unit of SARS-CoV-2 [22] (1:250) in 5% non-fat milk. After three washes with TBS-T, anti-Hamster IgG antibody conjugated to HRP (Abcam, USA) was added to the membrane at 1:5000 dilution in 5% non-fat milk and incubated for two hours at RT. Finally, the membranes were washed three times with TBST-0.1%, incubated with luminol (Azure Biosystems, USA) as a substrate and revealed with a CCD camera (Azure Biosystems, USA).

## Sodium dodecyl sulfate polyacrylamide gel electrophoresis (SDS-PAGE)

Protein samples were mixed with Laemmli 5x sample buffer in either reducing or non-reducing conditions and heated at 95°C for 5 minutes. Then, 20 μL of the sample were loaded to a 4–20% polyacrylamide gel (GensCript Laboratories, USA) and separated by electrophoresis at 100 V. Finally, gels were stained with Coomassie blue overnight at RT and unstained with a acetic acid:methanol:water (1:3:6) solution.

## RBD detection by Western blot

Supernatants of cells culture infected with the baculovirus expressing the RBD or a wild type baculovirus were electrophoretically separated in a 4–20% polyacrylamide gel under reducing conditions. Then, the proteins were transferred to a 0.22 μM nitrocellulose membrane using an E-blot L1 device (GenScript Laboratories, USA). The membranes were blocked with 5% (w/v) non-fat milk in PBS with 0.1% of Tween 20 for 1 hour at RT and washed three times with TBS-T for 5 minutes. Then, an anti His monoclonal antibody (GenScript Laboratiories, USA) or an anti-spike polyclonal antibody (SinoBiological, China) were added, both at a 2:5000 concentration in 5% (w/v) non-fat milk. After three washes steps, secondary antibodies anti-Mouse IgG antibody (1:5000) or anti-Rabbit IgG antibody (2:5000) conjugated to HRP were added. Finally, the membranes were washed three times with TBST-0.1%, incubated with luminol (Azure Biosystems, USA) as a substrate and revealed with a CCD camera (Azure Biosystems, USA)

## Immunization and samples collection in mice

Female BALB/c mice (18–25 g) were immunized intramuscularly (i.m.) with 20 or 50 μg/mice of purified RBD mixed with 50 μL of A1 or A3 (1:1, 100 μL final volume). Two boosters were administered at 15 and 30 days post-immunization (DPI) with the same dose (Fig 1). As a control, mice were immunized with PBS mixed with A1 or A3, an unvaccinated group was maintained during the experiment. Serum of each animal was collected on 0, 15, 30 and 45 DPI by low-speed centrifugation of blood at 2500 rpm for 5 minutes. All animals were euthanized at 45 DPI and organs (lung, liver and kidney) were collected for histopathological analysis.

## Immunization and samples collection in hamsters

Five Golden Syrian hamsters were immunized intramuscularly, each one with 30 μg of purified RBD mixed with oil adjuvant A3 (1:1) (which was the best adjuvant tested in mice) in a final volume of 100 μL (Fig 2). Five animals received only adjuvant A3 and were considered as the control group. At 15 DPI, all groups received a booster at the same dose. Hamsters were bled at 0, 15 and 30 DPI to evaluate the specific and neutralizing antibody (nAbs) titers. Serum from each sample was obtained by centrifugation of blood at 2500 rpm for 5 min.

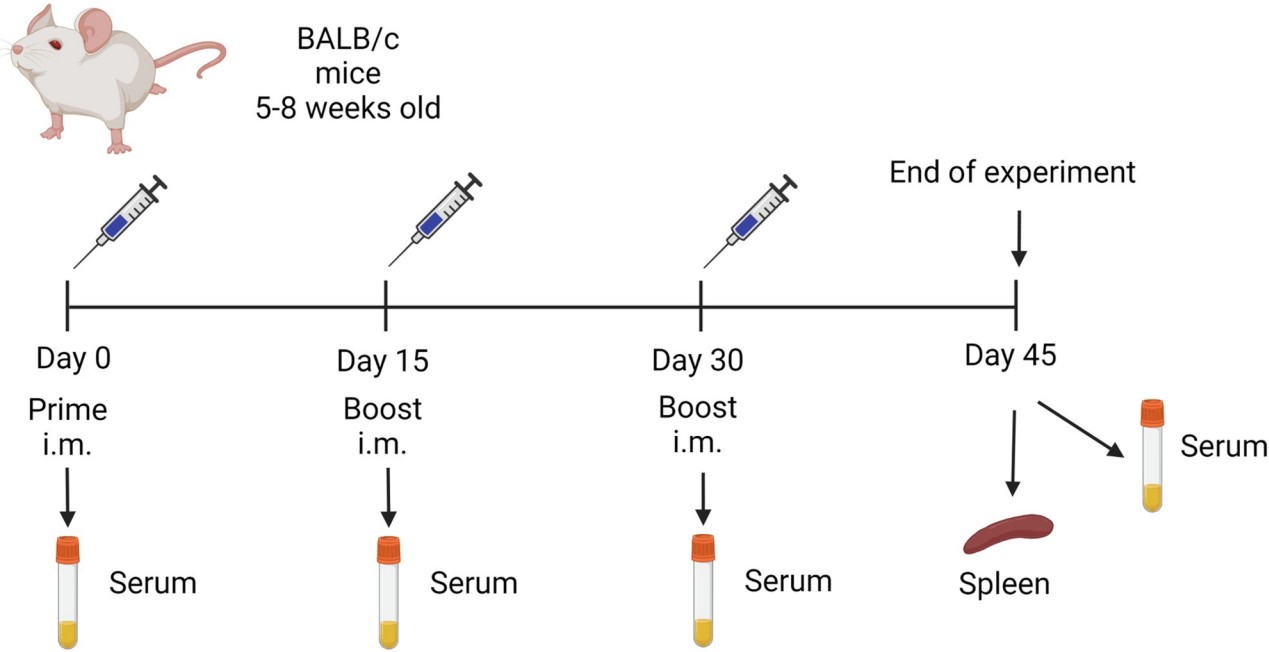

**Fig 1. Mice immunization flow chart.** Mice were immunized by the intramuscular route using a prime-boost regimen with a booster on days 15 and 30. Seven groups of mice were included: group 1 (20 µg RBD/A1, n = 5), group 2 (50 µg RBD/A1, n = 5), group 3 (20 µg RBD/A3, n = 5), group 4 (50 µg RBD/A3, n = 5), group 5 (only A1 n = 5), group 6 (only A3, n = 5) and group 7 (no immunization).

## Evaluation of humoral immunity

**Detection of specific antibodies by ELISA.** Nunc MaxiSorp 96-well flat bottom plates (Sigma-Aldrich, USA) were coated with 100 µL of SARS-CoV-2 RBD (1 µg/mL) (GenScript Laboratories, USA) in carbonate bicarbonate buffer (pH 9.6) and incubated at 4°C overnight. The next day, the wells were washed six times with PBS containing 0.05% (v/v) Tween-20 (PBS-T) and blocked with 3% (w/v) skim milk (BD Biosciences, USA) in PBS-T for 2 hours in agitation at RT. The plates were then washed six times with PBS-T. Then, 100 µL of each collected serum sample diluted 1:100 with 1% (w/v) skim milk was added to each plate for 1 hour at 37°C. The wells were washed six times with PBS-T and incubated with 100 µL (1:10000) of Goat Anti Mouse IgG (Genscript Laboratories, USA) or Anti Hamster IgG (Abcam, USA) conjugated to HRP diluted in skim milk in PBS-T for 1 hour at 37°C. The plates were washed six times and were incubated with 100 µL of TMB for 15 min at RT. Finally, the reaction was stopped by adding 50 µL per well of 2 N $H_2SO_4$, and the plates were read at 450 nm using an Epoch 2 microplate reader (Biotek, USA). The negative control was obtained from serum samples of the control group.

**Detection of neutralizing antibodies.** Hamster serum samples were processed to assess neutralizing antibodies (nAbs) against SARS-CoV-2 at 0, 15, and 30 days post immunization. All Neutralization assays were performed with the surrogate virus neutralization test (sVNT) (GenScript Laboratories, USA), following the manufacturer's instructions. Plates were read for absorbance at 450 nm using an Epoch 2 microplate reader (Biotek, USA). The optical density results were converted into percentage of inhibition, by the formula provided by the manufacturer. The positive and negative cut-off points for the detection of SARS-CoV-2 nAbs were set as follows: positive, if percentage of inhibition $\geq$ 30% (neutralizing antibody detected) and negative, if percentage of inhibition <30% (neutralizing antibody not detectable).

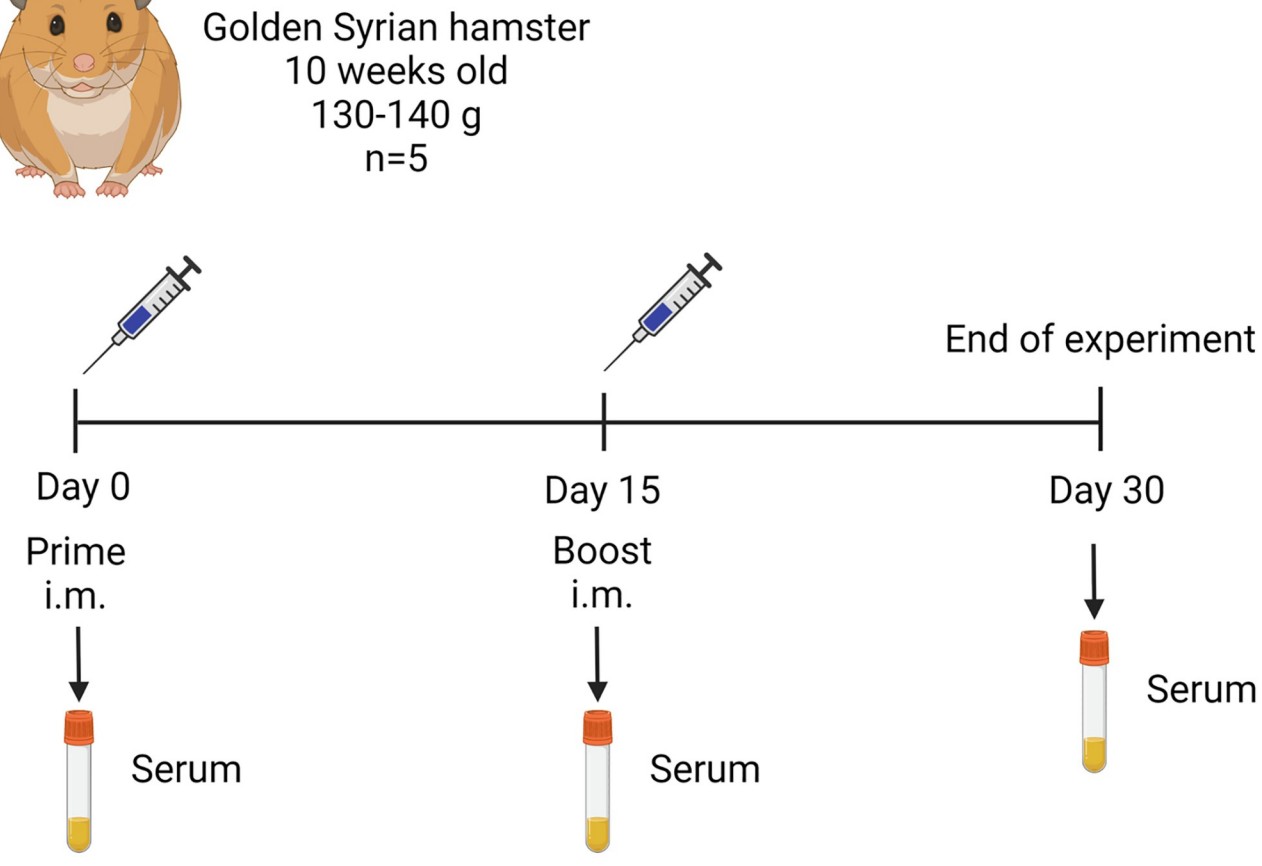

**Fig 2. Hamster immunization flow chart.** Hamsters were immunized by the intramuscular route with 30 μg of purified RBD in adjuvant A3 using a prime-boost regimen with a booster on day 15.

### Evaluation of cellular immunity

**Extraction of mononuclear cells from mouse spleen.** The mice vaccinated with the purified RBD and the control group (adjuvant only) were euthanized at 45 days post immunization, and spleens were removed. The organs were transferred to Petri dishes with 5 mL cold RPMI medium (Sigma Aldrich, USA) and two pieces of 41 μm nylon net (Merck, USA), where the organ was disrupted using a 3 mL syringe plunger. The cell suspension was filtered and placed in a centrifuge tube containing 2 mL of Histopaque® 1077 (Sigma Aldrich, USA). The samples were centrifuged at 300 x g for 30 min without brake. The buffy coat containing mononuclear cells was removed, placed in cold RPMI medium, and washed twice. Cells were resuspended in 1 mL of complete RPMI medium and counted by hemocytometer. Cells were resuspended in fetal bovine serum (HyClone, USA) with 10% dimethyl sulfoxide (Sigma Aldrich, USA) and frozen in liquid nitrogen until use.

**ELISPOT for IFN-γ secretion in spleen mononuclear cells.** Mononuclear cells were cultured in 96-well plates with a PVDF membrane, previously coated with anti-mouse IFN-γ (clone RMMG-1, Merck, USA) and blocked with 1% bovine serum albumin (BSA) (Sigma Aldrich, USA). Cells were stimulated with the purified RBD (4 μg/mL) for 24 hours at 37°C at 5% $CO_2$. Concanavalin A (Sigma Aldrich, USA) was used as a positive control. The cells were

removed by successive washes with water and PBS with 0.1% Tween. The wells were incubated with biotinylated anti-mouse IFN-γ (clone R4-6A2, Biolegend, USA) for 16 hours at 4°C. After washing, the wells were incubated with streptavidin-alkaline phosphatase (SAP) (Sigma Aldrich, USA) for one hour at RT and washed again. Then, the chromogen-substrate, NBT/BCIP (Abcam, USA), was added. The spots formed were counted with an AID EliSpot plate reader (Advanced Imaging Devices, v. 7.0, Germany).

**Intracellular staining of cellular immune response cytokines.** The mononuclear cells were stimulated with or without purified RBD (8 μg/mL) for 21 hours at 37°C at 5% $CO_2$. In the last 5 hours of culture a protein transport inhibitor, Brefeldin A, (1μL/mL) was added (BD Biosciences, USA). Cells were fixed using the BD Cytofix/Cytoperm® kit (BD Biosciences, USA) following the manufacturer's instructions, and then labeled with conjugated antibodies to surface antigens (PerCP-Cy®5.5 anti-mouse CD3, FITC anti-mouse CD4, APC-Cy®7 mouse anti-CD8, all from BD Biosciences, USA; LIVE/DEAD™ Fixable Yellow Dead Cell Stain, Invitrogen, USA) and intracellular cytokines (PE anti-mouse IFN-γ, PE-Cy®7 anti-mouse TNF-α, APC anti-mouse IL-2, all from BD Biosciences, USA). The labeled cells were acquired with the BD FACSCanto™ II flow cytometer and analyzed with the program FlowJo v.10.6.2 (BD Biosciences).

**Immunophenotype of spleen mononuclear cells.** Mononuclear cells were directly labeled with conjugated antibodies to surface antigens (PerCP-Cy®5.5 anti-mouse CD3, clone, FITC anti-mouse CD4, APC-Cy®7 anti-mouse CD8), for T lymphocyte phenotype, all from BD Biosciences, USA and LIVE/DEAD™ Fixable Yellow Dead Cell Stain, for cell viability (cat. No. L34959, invitrogen, USA). These cells were acquired with the BD FACSCanto™ II flow cytometer, and the analysis was performed with the program FlowJo v 10.6.2 (BD Biosciences).

## Histopathological analysis

For safety analysis, animals were anesthetized with 100 μL of Ketamine (100 mg), Xylazine (20 mg), and Atropine Sulfate (1 mg) via intramuscular (i.m.) injection and euthanized, organs were fixed with 10% buffered formalin for 48 hours. Then, organs were reduced and placed in a container for 24 hours with buffered formalin. The containers with the organs were passed to an automatic tissue processor (Microm brand) conducting the following processes: dehydration, diaphanating, rinsing, and impregnation; within an average of 8 hours. Organs included in paraffin were sectioned to a thickness of 5 microns (Microtome Leica RM2245) and placed in a flotation solution in a water bath and then fixed on a slide sheet, dried in the stove at 37°C for 1 to 2 hours. The staining was done with the Hematoxylin and Eosin staining method (H&E). Samples were mounted in a microscope slide with Canada Balm (glue) and dried at 37°C for 12 to 24 hours, for further labeling. The colored slides with H&E were taken and analyzed under an AxioCam MRc5 camera and AxioScope.A1 microscope (Carl Zeiss, Germany) at 20x magnification by a board-certified veterinary pathologist.

## Statistical analysis

All quantitative data were analyzed using GraphPad Prism version 6.1 (GraphPad Software, San Diego, CA, USA). Student t-test was used to evaluate cellular immunity. For EC50 estimation, a regression model of four parameters logistic curve (4PL) was used. Two-way ANOVA analysis was performed to determine significant difference in ELISA results. A 5% statistical significance was considered in all cases.

## Results

### Recombinant SARS-CoV-2 RBD production

Recombinant RBD was expressed and secreted into the extracellular medium by infected Sf-9 cells. A double band of ~28kDa was detected by western blot using Anti-his and Anti-spike antibodies (Fig 3B). In bioreactor conditions, the highest protein expression level was observed at 68 hours post-infection and after the purification processes, a productivity level of 0.8 mg/mL of RBD was obtained at a purity level > 90% (Fig 3C).

### Recombinant SARS-CoV-2 RBD characterization

To determine the correct conformational state of RBD, ACE-2 receptor binding assays were performed. ACE-2 binding dependent on RBD concentration was observed, with a half maximal effective concentration (EC50) of 46.8 ng/mL (Fig 4A). Similarly, through flow cytometry, RBD bound to Vero E6 cell surface at different concentrations, with a 60% binding level (Fig

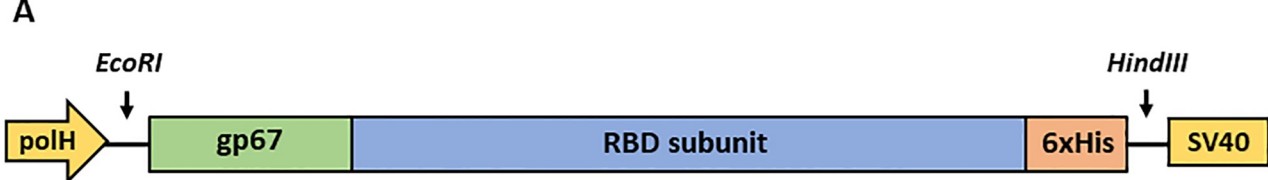

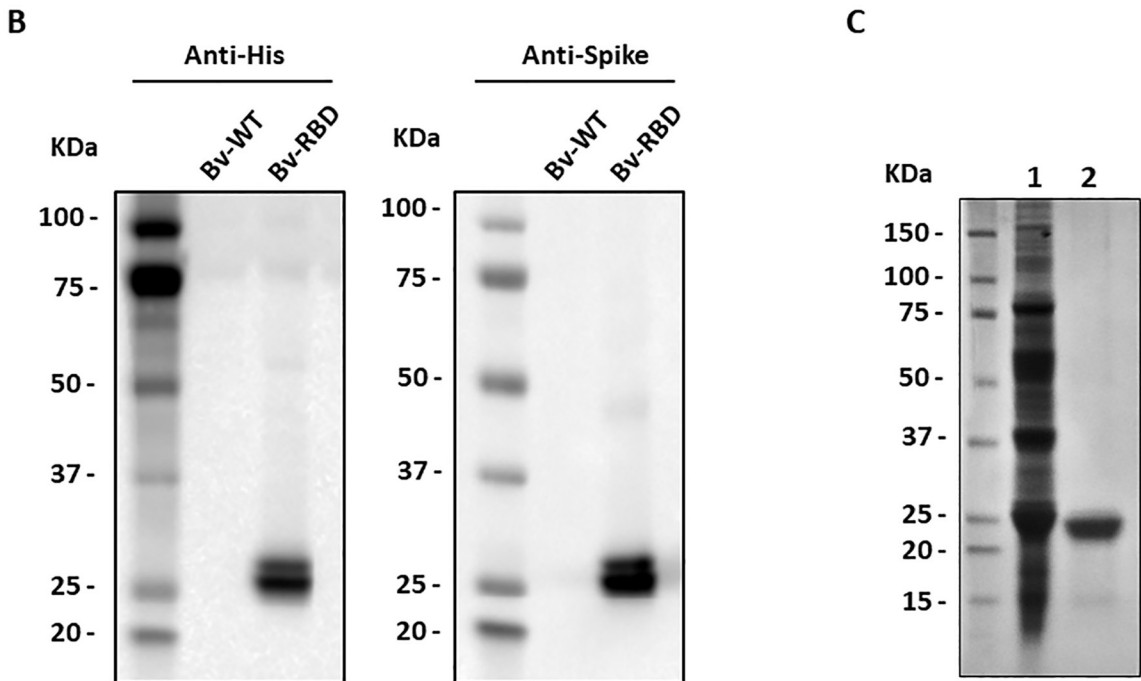

**Fig 3. RBD expression and purification.** (A) Design of the expression cassette integrated into the recombinant baculovirus. (B) Detection of RBD from infected culture supernatants using an anti-His (left) and anti-spike (right) antibody. Bv-WT: Wild type baculovirus; Bv-RBD: RBD expressing baculovirus. (C) SDS-PAGE of purified RBD after the affinity chromatography purification step (Lane 1) and size exclusion chromatography (Lane 2).

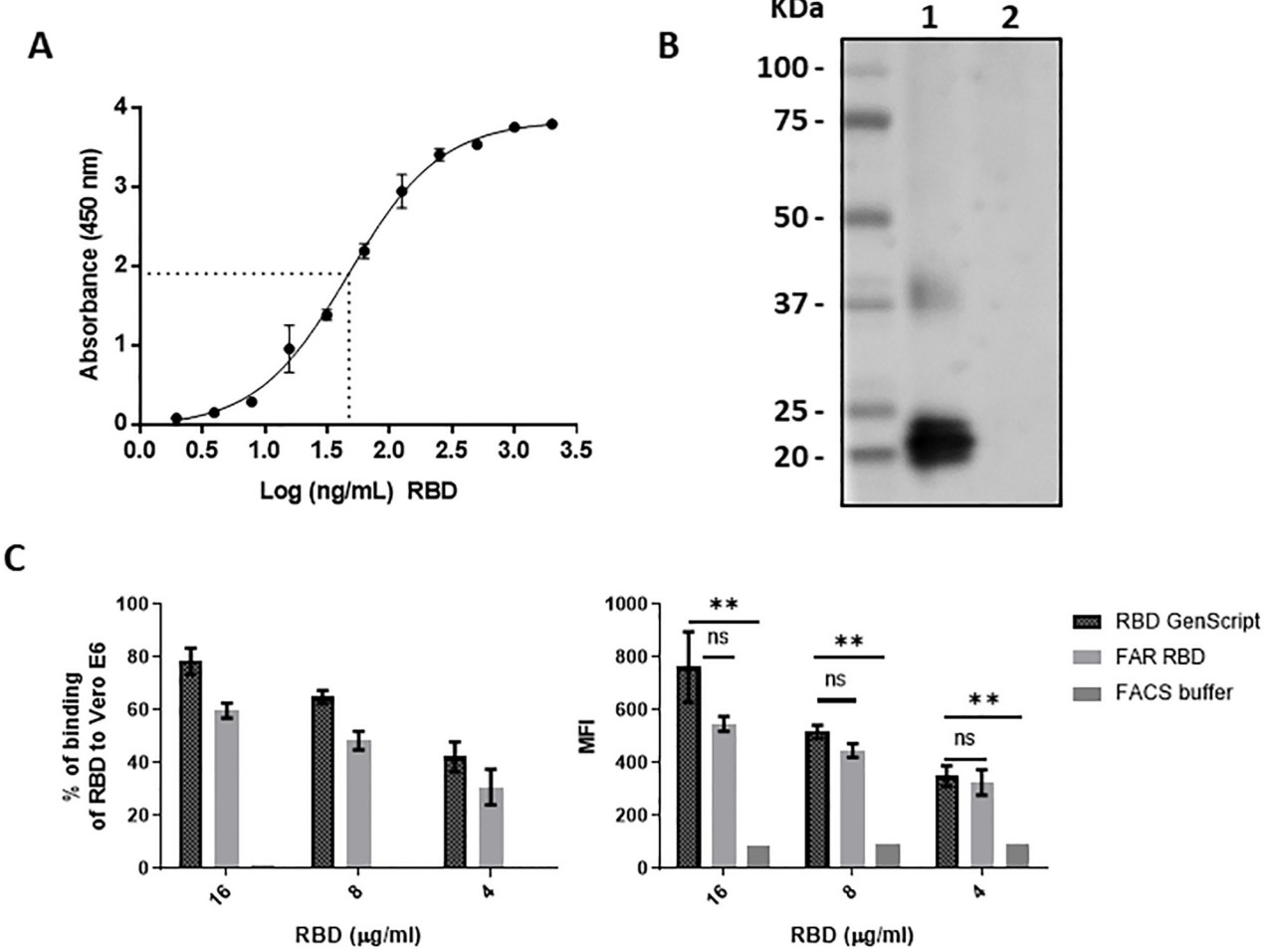

**Fig 4. RBD binding and folding characterization in vitro.** (A) Dose dependent curve of RBD binding to human ACE-2 by ELISA, dashed lines represent the EC50 value. Dots and error bars represent the mean value of three independent experiments and the standard deviation, respectively. (B) Disulfide bond dependent recognition of RBD by hamsters immunized serum by Western blot. Lane1: RBD under non-reducing conditions; Lane 2: RBD under reducing conditions. (C) RBD binding to Vero E6 cell surface. The binding values are represented as the percentage of cells bound to RBD (left diagram) and the Mean Fluorescence Intensity (MFI) of each group was evaluated (right diagram). Two repetitions were performed per group, except in the FACS buffer group. Student $t$-test was used to compare the MFI values. ns: not significant (P>0.05); **: significant (P<0.01).

4C). Based on the main fluorescence intensity (MFI), the difference between the cells treated with purified RBD and those treated with FACS buffer as a negative control was significant. On the other hand, commercially available recombinant RBD expressed in insect cells (Cat No. Z03479, GenScript) was used as a positive control, although its binding was slightly higher than the RBD produced in this study (78%), the difference was not statistically significant. This trend was observed in all the concentrations evaluated.

The importance of disulfide bonds for the correct folding of the RBD sub-domain is known. Therefore, an additional way to verify the correct folding of the recombinant RBD was evaluating its detection under reducing and non-reducing conditions, by using a serum from a hamster immunized with a New Castle Disease virus (NDV) expressing the S1 domain (Fig 4B). In this way by Western blot, RBD could be detected by the serum only under non-reducing conditions, demonstrating that it conserves the folding of the RBD sub-domain occurring in the Spike protein.

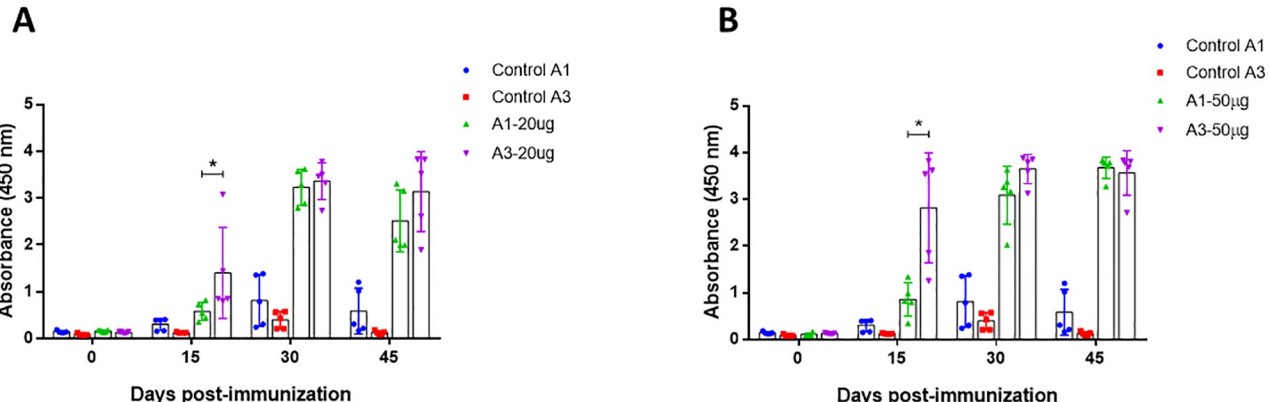

**Fig 5. Detection of specific antibodies against RBD in mice.** Immunized mice were bled at 0, 15, 30 and 45 days post immunization. All sera were obtained by low-speed centrifugation. Serum samples were processed to detect specific antibodies against SARS-CoV-2 RBD protein using indirect ELISA assay. (A) Group immunized with 20 μg of RBD mixed with A1 and A3 (B) Group immunized with RBD 50 μg of RBD mixed with A1 and A4. Two-way ANOVA and post-hoc Tukey's test were performed. *: P<0.01.

## Humoral immunity

In order to compare the capacity of both oil adjuvants to enhance the immune response, two amounts of RBD were administered with each adjuvant in mice. Specific antibodies were detected in all immunized groups, at 15 days post immunization, antibody levels of the group immunized with adjuvant 3 were higher than the group with adjuvant 1, either with the 20 μg and 50 μg dose. However, after the first booster the levels of antibodies generated with both adjuvants were similar in the two doses of RBD evaluated (Fig 5), indicating that the early generation of antibodies in A3 respect to A1 was independent of the dose of protein administered. Control groups immunized with each adjuvant and PBS had baseline reactivity throughout the evaluation time.

Since with adjuvant 3, a stronger immune response was obtained in less time and with a single boost, this adjuvant was used to immunize hamsters. In this way, a significant increase in specific antibody levels was observed from day 15 post-immunization until day 30 in all the individuals tested (Fig 6A). The neutralization assays using the surrogate virus neutralization test (sVNT) detected neutralizing antibodies only at day 30 post-immunization, where the sera from hamsters vaccinated showed a mean percentage of inhibition of the RBD-ACE2 union above 30%. Sera of the control group remained below 30% and did not show neutralizing antibodies (Fig 6B).

## Cellular immunity

The cellular immunity stimulated with the purified RBD in mice was evaluated on day 45 after the first immunization. For adjuvant A1, the percentage of CD4+ and CD8+ T cells increased proportional to the dose of RBD administered. However, in A3 group the percentage of cells decreased when the dose of RBD was increased (Fig 7A). Regarding the production of Th1-type cytokines (Fig 7C), the number of CD8+ T cells secreting IFN-γ, TNF-α and IL-2 for A1 and A3 groups decreased with the highest dose of RBD. However, CD4+ T cells producing these Th1-type cytokines were not detected in all groups. Regarding the secretion of IFN-γ in splenocytes stimulated with purified RBD using the ELISPOT technique (Fig 7B), the adjuvant A3 stimulated a greater number of cells directly proportional to the administered dose. Although there was difference in the mean values of each group, when the statistical analysis

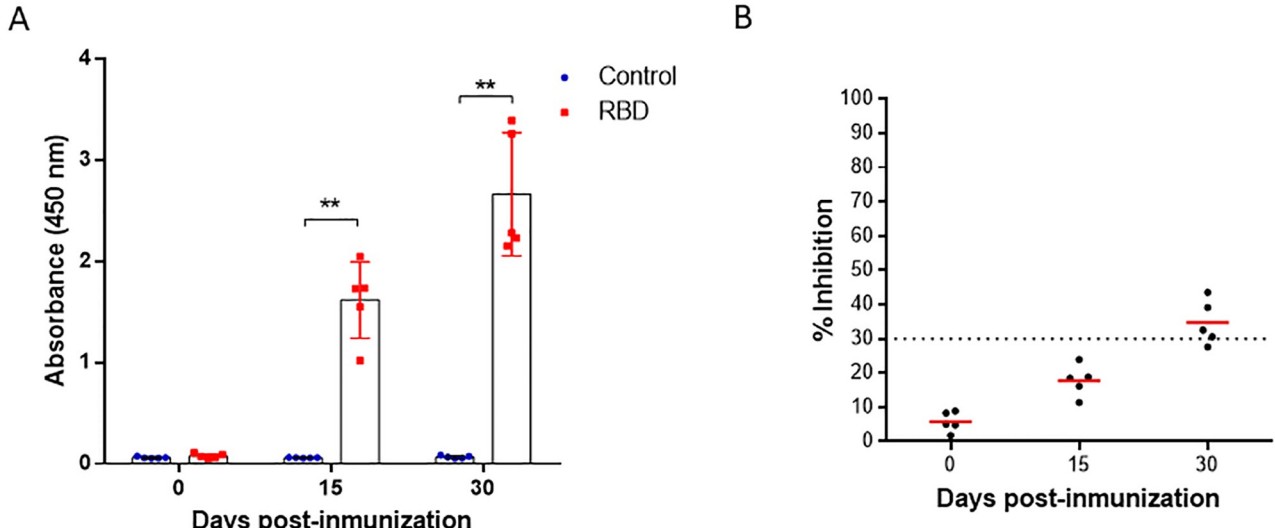

**Fig 6. Detection of specific antibodies against RBD and neutralizing antibodies in hamsters.** (A) Immunized hamsters were bled at 0, 15 and 30 days post immunization. Serum samples were processed to detect specific antibodies against SARS-CoV-2 RBD protein using indirect ELISA assay. (B) Serum samples were processed to evaluate the neutralizing antibody titers against SARS-CoV-2 using sVNT. The cut-off for positive/negative neutralizing antibodies in the sample was 30% of inhibition of RBD binding to ACE-2. Two-way ANOVA and post-hoc Tukey's test were performed. **: $P < 0.0001$.

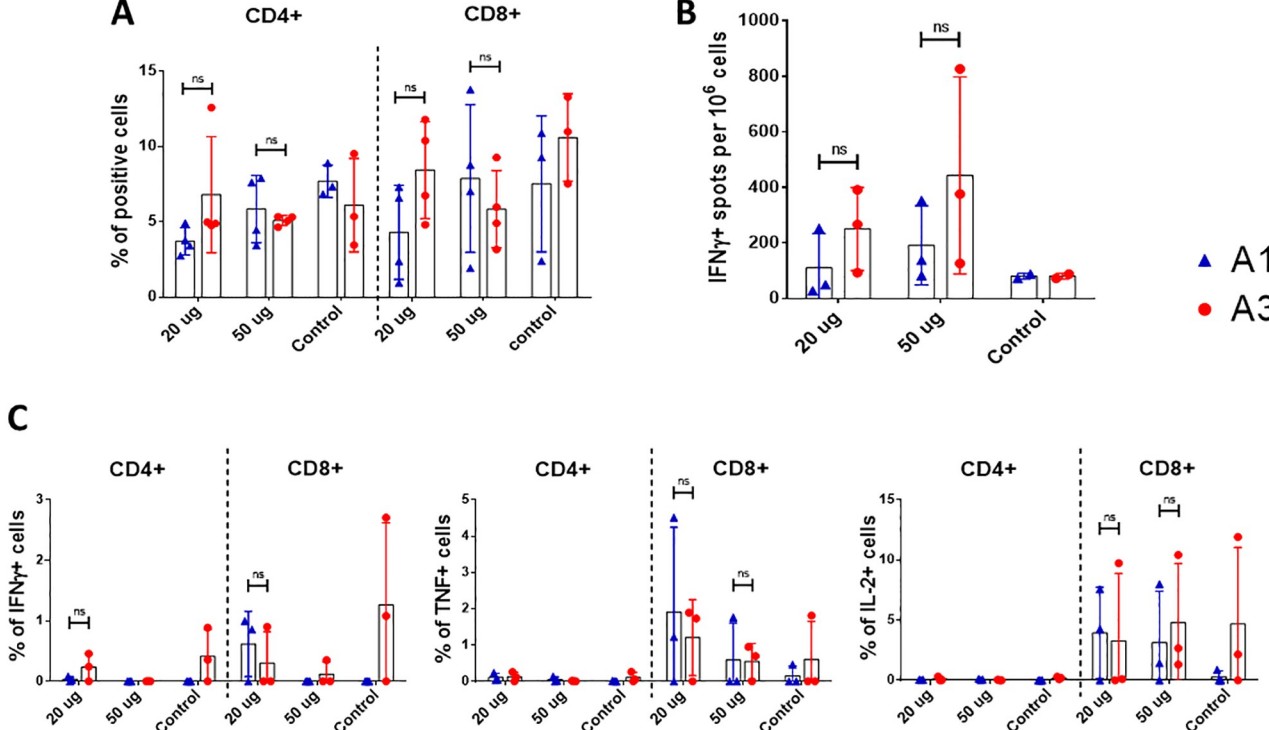

**Fig 7. Evaluation of cellular immunity in mice vaccinated with purified RBD.** Mice were immunized with 20 and 50 μg of RBD using two different adjuvants (A1 and A3) at 0, 15 and 30 days post immunization. On day 45 post-immunization mice were euthanized and spleens were processed. (A) Percentage of CD4 and CD8 positive cells by flow cytometry, between the groups immunized (n = 3, except the adjuvant control). (B) IFN-γ ELISPOT of splenocytes between the groups immunized (n = 3, except the adjuvant control). (C) Intracellular staining of Th1 cytokines (IFN-γ, TNF-α and IL-2) of splenocytes stimulated with RBD (n = 3, except the adjuvant control). ns: not significant ($P > 0.05$), *: $P < 0.05$.

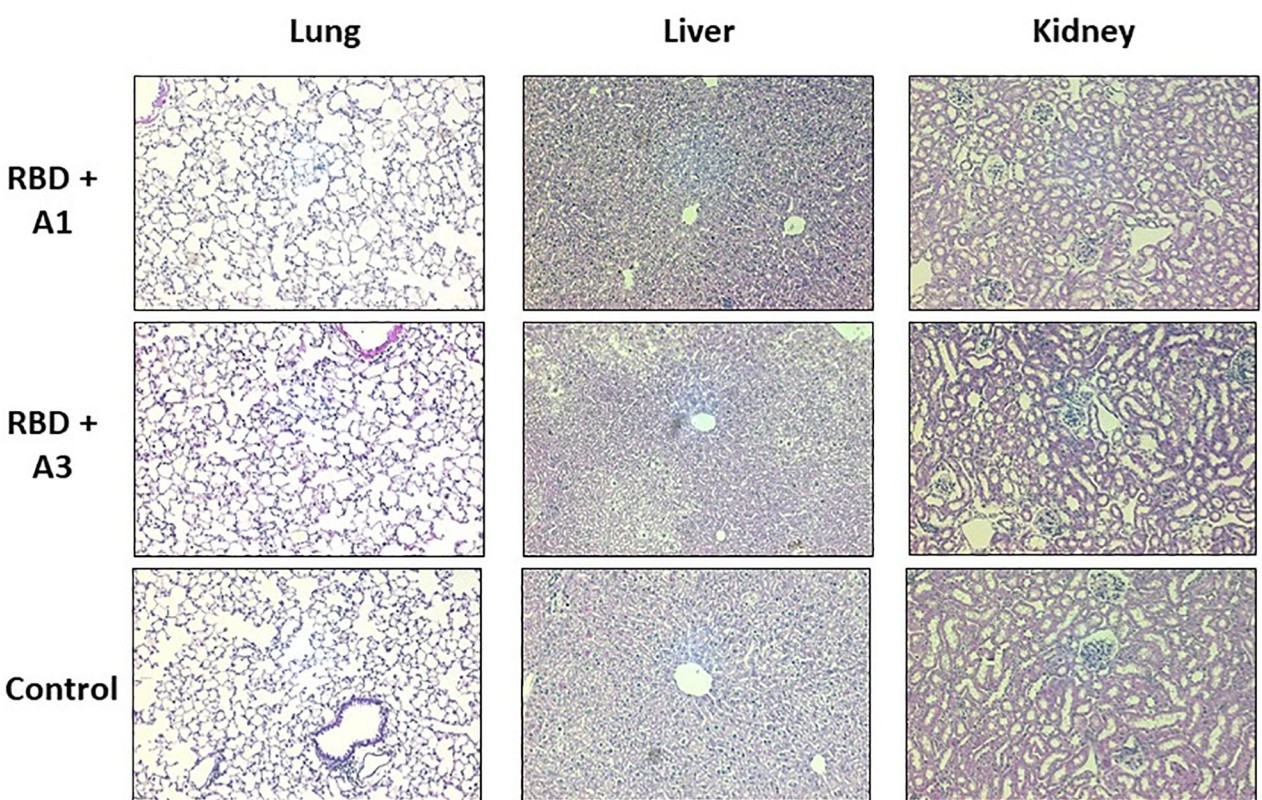

**Fig 8. Histopathological analysis of mice inoculated with purified RBD and control.** Organs were obtained 45 days after the first immunization and stained with hematoxylin-eosin (H&E). These images are representative slides from vaccinated mice and negative control mice. (A) Lung sections. (B) Liver sections. (C) Kidney sections. All the images are in a 200X magnification.

was performed there was not significant difference between them in all the evaluations performed.

## Safety

Histopathological analysis of the groups of mice immunized with purified RBD mixed with A1 or A3, including the unvaccinated group not showed signs of serious injury or damage. Lungs not showed clinical appearance of pneumonia and there was no evidence of kidney symptoms. Although in liver a slight vacuolar degeneration was identified, this was observed in all the groups tested, including the control group. (Fig 8).

## Discussion

SARS-CoV-2 continues to be a problem worldwide. As an immediate response to the emergence of new variants and their dissemination, the constant development and evaluation of vaccines are necessary. In the present study, the immunization of the RBD sub-domain mixed with two different oil-based adjuvants demonstrate that squalene improves the immunogenicity by eliciting an earlier humoral response in mice and hamster.

Currently, most of the approved and candidate vaccines are based on the complete spike protein. However, there are several vaccine candidates based on the single RBD antigen, ongoing pre-clinical and clinical phase [23]. Although, in some reports the complete spike has

shown greater immunogenicity [24], the single RBD remains as a strong vaccine candidate because it comprises the most important epitopes to which neutralizing antibodies should target. In addition, it generates antibodies with enhanced neutralizing activity [25–27] and the greater accumulation of mutations in the S1 and S2 domains can destabilize the protein, hindering its production and the yields obtained as a purified protein [28, 29]. On the other hand, RBD has demonstrated an easier production [30], and results in a more conserved antigen. Recent studies have bioengineered RBD variants with improved stability and higher immune response in mice compared to the current Wuhan-Hu-1 vaccine [31]. Likewise, a thermotolerant RBD fused to a trimerization motif has generated high neutralization titers in guinea pigs and mice, as well as protection in hamsters from viral challenge [32].

Despite that the purified RBD evaluated in this study comprised 22 amino acids less than the generally recognized RBD region (Arg319-Phe541) [33], it was structurally and functionally viable as demonstrated by the binding assays by ELISA and flow cytometry. Although a double band was observed after the purification process, they corresponded to the RBD since both were recognized by all the antibodies used. This double band could be explained to a difference in glycosylation patterns in the RBD, as this phenomenon has been reported in the expression of other proteins in insect cells [12, 34]. Apparently, this possible difference in glycosylation patterns do not alter the function or the structure of the RBD, since the final EC50 value of binding to human ACE-2 was comparable with previous reports [4]. This functionality was maintained because the expressed region comprises the residues that form the disulfide bonds that give stability to the nucleus and the key external sub-domains of the RBD [3], maintaining the integrity of the receptor binding motif, which ultimately is the main region that directly interacts with the ACE2 receptor. This was confirmed by the lack of RBD recognition of the hamster anti-S1 immune sera under reducing conditions, but the strong recognition of RBD by the immune sera under non-reducing conditions. This suggests, that the disulfide bonds are present and are favoring a correct folding and 3D structure of the RBD antigen, that may be presenting appropriate conformational epitopes, as most of the immune antibodies targets tertiary epitopes spanning the exposed sites of the RBD in the trimeric pre-fusion Spike [35].

The production level of RBD in this study, was relatively low (0.8 mg/L) compared to previous reports of expression of the same domain using the baculovirus expression system [30]. It is likely that this is due to the baculovirus type used, which is not optimized for secreted expression, or to the second purification step required to obtain a higher degree of purity. These levels could be optimized using baculoviruses lacking the *v-cath* and *chiA* genes [36] or through optimization strategies of the amino acid sequence that have been proven to improve expression levels and immunogenicity of the RBD [31, 37].

The chemical composition of an adjuvant is important because its components may interfere with organism responses. In the immunization experiments conducted in this study, adjuvant 3 (W/O + squalene) was associated to higher levels of anti-RBD antibodies than adjuvant 1 (O/W) at 15 days post immunization. However, after the second booster was administered (45 DPI), the antibody levels for both adjuvants were not significantly different. This could be explained by the fact that O/W emulsions, as an adjuvant for mice, generates higher levels of antibodies while directing the cellular immune response to the Th2 type [38]. Also, it is known that O/W emulsions stimulate a strong production of TNF-α [16, 39] and do not generate local inflammation reactions when injected subcutaneously or intramuscularly [40]. On the other hand, the W/O adjuvant formulations are not effective enough to induce strong humoral responses, as they can generate inflammatory responses and the formation of granulomas [41]. In contrast, adjuvant A3, which is a novel composition, demonstrates the stimulation of an earlier strong humoral response. Most oil-water (O/W) adjuvants that contain squalene, also

have other components (Tween 80, Span 85, polyethylene glycol or derivatives), which when emulsified in an aqueous phase, generate a stable chemical structure that allows the transport of antigens for their recognition by cells such as macrophages or dendritic cells [18, 38]. We believe further studies are necessary to clarify and confirm these observations.

When adjuvant A1 was administered with the purified RBD, the formulation did not generate IFN-γ, IL-2 nor TNF-α in the evaluation by ICS. However, an increase in the percentage of CD4+ and CD8+ T cells was observed. This observation is in agreement with a previous study, where Arunachalam et al. [42] found that adjuvant A1 (Essai O/W 1849101, Seppic) added to RBD nanoparticles did not elicit a strong antibody response nor protection as expected in Rhesus monkeys. Nevertheless, when A1 was used with alpha-tocopherol it produced a stronger level of neutralizing antibodies and protection against infection with SARS-CoV-2. However, the use of this adjuvant generated an inflammatory response, associated with a high expression of TNF-α and IL-2. We found that the novel adjuvant A3 stimulated the secretion of greater IFN-γ levels in splenocytes compared to adjuvant A1, as well as IL-2 and TNF-α in CD8+ T cells. This is consistent with the possible inflammatory effect generated by adjuvants based on W/O emulsions [41].

The generation of neutralizing antibodies in hamsters was observed at 30 days post immunization. Although the surrogate test does not directly determine the neutralization of virus invasiveness in cells, it has been shown that it has a high correlation index with classic viral neutralization tests [8]. In addition, various studies have demonstrated a relationship between the development of neutralizing antibodies with the protection of re-infection in humans, as well as in challenge tests in hamsters [43, 44].

Due to limitations in space and the availability of animals, this trial was conducted with 5 individuals per group, and the heterogeneity was evident as previously reported in a similar protocol [45]. Unfortunately, it was not possible to establish clear conclusions about the tendency of the population when stimulated with the two different adjuvants, as there was no significant difference between the controls and the immunized groups. It is important to perform additional studies with a greater sample size to perform a better evaluation of cellular and humoral immunity [13, 16, 26] to 8 per group as in previous studies [46].

In conclusion, the use of squalene in an oil-based adjuvant enhanced the immunogenicity of the RBD of SARS-CoV-2, this by stimulating an earlier generation of a humoral immunity and confirming its safety in mice. However, further studies are required to evaluate protection in a challenge trial.

## Supporting information

**S1 Fig. Gating strategy to determine the recombinant RBD binding to Vero-E6 cells.**
(TIF)

**S2 Fig. Resulting plots from the incubation of Vero E6 cells with different concentrations of recombinant RBD.** Two replicates were performed for each RBD evaluated.
(TIF)

**S3 Fig. Gating strategy for intracellular cytokine staining (ICS) and immunophenotype of mice spleen cells.**
(TIF)

**S4 Fig. Representative plots of CD4+ and CD8+ cells from each group.** Groups immunized with adjuvant 1 (A, B). Groups immunized with adjuvant 3 (C, D).
(TIF)

**S1 Dataset. Raw values of ELISA and Flow cytometry results.**
(ZIP)

**S1 Raw images.**
(PDF)

## Acknowledgments

We acknowledge Katherine Calderón, Aldo Rojas, Naer Chipana-Flores, Elmer Delgado, Abraham Licla, Katherine Pauyac, Luis Tataje and Julio Ticona from Laboratorios de investigación y desarrollo, FARVET SAC and Ricardo Antiparra, Manuel Ardiles, Yudith Cauna, Xiomara Chunga, Lewis De La Cruz, Nicolas Delgado, Christian Elugo, Oscar Heredia, Pedro Huerta, Grabriel Jiménez, Romina Juscamaita, Dennis Nuñez,, Adiana Ochoa, Gustavo Olivos, Erika Páucar, Jose Perez, Daniel Ramos, Angela Rios, Mario Salguedo, Patricia Sheen, Luis Soto, Anda Vargas and Renzo Villanela from Laboratorio de Bioinformática, Biología Molecular y Desarrollos Tecnológicos. Laboratorios de Investigación y Desarrollo. Facultad de Ciencias y Filosofía. Universidad Peruana Cayetano Heredia. All the names listed are members of the COVID-19 Working Group in Perú, whose author is Mirko Zimic (mirko.zimic@upch.pe).

## Author Contributions

**Conceptualization:** Ricardo Choque-Guevara, Stefany Quiñones-Garcia, Katherine Vallejos-Sánchez, Manolo Fernández-Sánchez, Luis A. Guevara-Sarmiento, Manolo Fernández-Díaz, Mirko Zimic.

**Data curation:** Ricardo Choque-Guevara, Astrid Poma-Acevedo, Ricardo Montesinos-Millán, Dora Rios-Matos, Angela Montalvan-Avalos, Stefany Quiñones-Garcia, Maria de Grecia Cauti-Mendoza, Andres Agurto-Arteaga, Ingrid Ramirez-Ortiz.

**Formal analysis:** Ricardo Choque-Guevara, Astrid Poma-Acevedo, Ricardo Montesinos-Millán, Dora Rios-Matos, Stefany Quiñones-Garcia, Maria de Grecia Cauti-Mendoza, Andres Agurto-Arteaga, Yomara K. Romero, Norma Perez-Martinez, Gisela Isasi-Rivas, Freddy Ygnacio, Manolo Fernández-Díaz.

**Funding acquisition:** Manolo Fernández-Sánchez, Manolo Fernández-Díaz, Mirko Zimic.

**Investigation:** Ricardo Choque-Guevara, Astrid Poma-Acevedo, Ricardo Montesinos-Millán, Dora Rios-Matos, Kristel Gutiérrez-Manchay, Angela Montalvan-Avalos, Stefany Quiñones-Garcia, Maria de Grecia Cauti-Mendoza, Andres Agurto-Arteaga, Ingrid Ramirez-Ortiz, Manuel Criollo-Orozco, Edison Huaccachi-Gonzales, Yacory Sernaque-Aguilar, Doris Villanueva-Pérez, Katherine Vallejos-Sánchez.

**Methodology:** Ricardo Choque-Guevara, Astrid Poma-Acevedo, Ricardo Montesinos-Millán, Dora Rios-Matos, Kristel Gutiérrez-Manchay, Angela Montalvan-Avalos, Stefany Quiñones-Garcia, Maria de Grecia Cauti-Mendoza, Ingrid Ramirez-Ortiz, Manuel Criollo-Orozco, Edison Huaccachi-Gonzales, Yomara K. Romero, Norma Perez-Martinez, Gisela Isasi-Rivas, Yacory Sernaque-Aguilar, Doris Villanueva-Pérez, Katherine Vallejos-Sánchez.

**Project administration:** Ricardo Choque-Guevara, Manolo Fernández-Sánchez, Luis A. Guevara-Sarmiento, Manolo Fernández-Díaz, Mirko Zimic.

**Resources:** Dora Rios-Matos, Luis A. Guevara-Sarmiento, Mirko Zimic.

**Supervision:** Manolo Fernández-Sánchez, Manolo Fernández-Díaz, Mirko Zimic.

**Validation:** Astrid Poma-Acevedo, Kristel Gutiérrez-Manchay.

**Visualization:** Astrid Poma-Acevedo.

**Writing – original draft:** Ricardo Choque-Guevara, Astrid Poma-Acevedo, Ricardo Montesinos-Millán, Dora Rios-Matos.

**Writing – review & editing:** Ricardo Choque-Guevara, Astrid Poma-Acevedo, Ricardo Montesinos-Millán, Dora Rios-Matos, Maria de Grecia Cauti-Mendoza, Manolo Fernández-Sánchez, Manolo Fernández-Díaz, Mirko Zimic.

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
