## [Decision Letter · Decision Letter 0]

5 Jan 2022

PONE-D-21-36362A recombinant SARS-CoV-2 RBD antigen expressed in insect cells elicits immunogenicity and confirms safety in animal modelsPLOS ONE

Dear Dr. Zimic,

Thank you for submitting your manuscript to PLOS ONE. After careful consideration, we feel that it has merit but does not fully meet PLOS ONE’s publication criteria as it currently stands. Therefore, we invite you to submit a revised version of the manuscript that addresses the points raised during the review process.

We look forward to receiving your revised manuscript.

Kind regards,

Paulo Lee Ho, Ph.D.

Academic Editor

PLOS ONE

Journal Requirements:

2. To comply with PLOS ONE submissions requirements, in your Methods section, please provide additional information on the animal research and ensure you have included details on (1) methods of sacrifice, (2) methods of anesthesia and/or analgesia, and (3) efforts to alleviate suffering.

4. One of the noted authors is a group or consortium COVID-19 Working Group. In addition to naming the author group, please list the individual authors and affiliations within this group in the acknowledgments section of your manuscript. Please also indicate clearly a lead author for this group along with a contact email address.

Reviewers' comments:

Reviewer's Responses to Questions

**Comments to the Author**

1. Is the manuscript technically sound, and do the data support the conclusions?

Reviewer #1: Yes

Reviewer #2: Partly

2. Has the statistical analysis been performed appropriately and rigorously? 

Reviewer #1: Yes

Reviewer #2: Yes

3. Have the authors made all data underlying the findings in their manuscript fully available?

Reviewer #1: Yes

Reviewer #2: Yes

4. Is the manuscript presented in an intelligible fashion and written in standard English?

Reviewer #1: Yes

Reviewer #2: Yes

5. Review Comments to the Author

Reviewer #1: The authors describe the production, purification and immunological characterization of RBD obtained from insect cells. It is not clear in the article, how the quantification of RBD was carried out, nor the methods used to obtain the SDS-PAGE and Western blot, especially the specification of the anti-S antibodies used. Furthermore, in line 332, the authors describe the visualization of only one band around 28kDa, but on the gel (figure 3B) it is possible to visualize 2 bands corresponding to the labeling with the antibodies.

Regarding the in vivo tests, it is not clear why the hamster challenge test was not done. Even if the clinical symptoms of these animals infected with SARS CoV-2 are rapid and transient, it would be possible to obtain a proof of concept in relation to the studied formulations.

Reviewer #2: � Minor concerns:

• Authors should addressing the novelty in their work, for example (similar work, published in 2020, doi: 10.1080/22221751.2020.1821583), as the concept of using RBD or full spike have been published? Even some commercial approved subunit vaccines now in use (like: NOVAVAX).

• Why Authors didn’t test the recombinant purified RBD protein antigenicity against the convalescent COVID-19 patient?

• What is the source of RBD that used as a positive control in bounding with Vero-E6 cell?

Some examples for sentences need careful editorial review:

Some sentences in “Abstract” are too long and containing repetition: from line 38-42 (purified RBD mentioned two times in the same sentence and also the meaning of stimulation), please rewrite in a shorter way.

• Please unify the use the abbreviation of “W/O” with or without slash; hours (completed or abbreviated); minutes; temperature; room temperature;

• Please make “space” in line 51 “atypical”, line 55 “add for after urgent”

• “°C” with or without space; missing some ending dots like in line 158 after membrane; at the end of line 503.

• Remove the “dot” after then in line 309

• Eliminate the word “aseptically” in line 265

• The word “while” is misleading in line 499

6. PLOS authors have the option to publish the peer review history of their article (what does this mean?). If published, this will include your full peer review and any attached files.

Reviewer #1: No

Reviewer #2: **Yes: **Reda Salem

---

## [Author Response · Author response to Decision Letter 0]

19 Feb 2022

Editor Comments:

Response:

The indentation format has been included at the beginning of each paragraph as specified by the magazine's style requirements.

2. To comply with PLOS ONE submissions requirements, in your Methods section, please provide additional information on the animal research and ensure you have included details on (1) methods of sacrifice, (2) methods of anesthesia and/or analgesia, and (3) efforts to alleviate suffering.

Response:

The required information has been included in the ethical statement section in the materials and methods section, on lines 111 - 113 of the unmarked version of the modified manuscript:

"The animals were euthanized by trained veterinary personnel following the guideline established by the American Veterinary Medical Association (AVMA) [21]."

As well as in the Immunization and sample collection in mice, in lines 251 – 255:

“Briefly, mice were euthanized by anesthetic overdose, inoculating 200 µL of a ketamine (100 mg/mL), xylazine (20 mg/mL) and atropine (1 mg/mL) solution using a hypodermic needle by intramuscular route. The procedure was performed rapidly to minimize the suffering, the animal was kept in a quiet place until the effects of anesthesia began to manifest.”

Response:

Unfortunately, we had an error during submission of the manuscript. We are willing to make our results data available as Supplementary Material in an Excel file that can be made available by the reviewers as well as for final publication upon request. We would be grateful if the change of the data availability statement to the above-mentioned mode could be made.

4. One of the noted authors is a group or consortium COVID-19 Working Group. In addition to naming the author group, please list the individual authors and affiliations within this group in the acknowledgments section of your manuscript. Please also indicate clearly a lead author for this group along with a contact email address.

Response:

The list of the COVID-19 Working Group consortium has been included, as well as the author of the group with his contact e-mail address. This can be found in the acknowledgments section on lines 592 to 602 of the modified manuscript version.

Response:

We thank the editor for the comment. We have modified the manuscript and the ethical statement is included only in the materials and methods section, on lines 105 to 113 of the corrected version of the manuscript.

6. PLOS ONE now requires that authors provide the original uncropped and unadjusted images underlying all blot or gel results reported in a submission’s figures or Supporting Information files.

Response:

We have attached all the images in RAW format without editing, following the requirements for blot and gel submissions. You can see them in the S1_raw_images file attached along with the corrected manuscript.

Reviewers comments:

Reviewer 1

1. The authors describe the production, purification and immunological characterization of RBD obtained from insect cells. It is not clear in the article, how the quantification of RBD was carried out, nor the methods used to obtain the SDS-PAGE and Western blot, especially the specification of the anti-S antibodies used. Furthermore, in line 332, the authors describe the visualization of only one band around 28kDa, but on the gel (figure 3B) it is possible to visualize 2 bands corresponding to the labeling with the antibodies.

Response:

RBD quantification was performed by the standard estimation technique using Bradford's reagent, as described in lines 169-170 of the unmarked corrected manuscript. 

“The concentration of purified RBD was determined using the Bradford assay (Merck, Germany), following the manufacturer’s instructions.”

Regarding the SDS-PAGE and Western bot methodologies, this has been added in lines 221 - 242. In addition, this section specifies the anti-S antibodies used, which can be found in lines 235 - 237 of the same document 

“Then, an anti 6x-His monoclonal antibody (GenScript Laboratories, USA) or an anti-Spike polyclonal antibody (SinoBiological, China)”

Similarly, the description in the western blot results in Figure 3B has been modified. This can be found on line 379-380 of the modified manuscript. 

“A double band of ~28KDa was detected by western blot using Anti-his and Anti-spike antibodies (Fig 3B).”

This result has also been described in the discussion section on lines 519 - 525, where possible reasons for the purification of RBD as a double band are discussed. 

“Although a double band was observed after the purification process, they corresponded to the RBD since both were recognized by all the antibodies used. This double band could be explained by a difference in glycosylation patterns in the RBD, as this phenomenon has been reported in the expression of other proteins in insect cells [12,34]. This possible difference in glycosylation patterns does not alter the function or the structure of the RBD, since the final EC50 value of binding to human ACE-2 was comparable with previous reports [4]”

2. Regarding the in vivo tests, it is not clear why the hamster challenge test was not done. Even if the clinical symptoms of these animals infected with SARS CoV-2 are rapid and transient, it would be possible to obtain a proof of concept in relation to the studied formulations.

Response:

We understand that a challenge against the virus would have been positive for the research objectives. Unfortunately, during the execution of the study we had difficulties in obtaining facilities with the biosafety level to work with the virus, since our institution did not yet have the resources to build a biosafety level 3 (BSL3) facility. In addition, at the time of the study, our country's national health institute had not yet isolated a strain of SARS-CoV-2 for in vivo use.

Reviewer 2

1. Authors should addressing the novelty in their work, for example (similar work, published in 2020, doi: 10.1080/22221751.2020.1821583), as the concept of using RBD or full spike have been published? Even some commercial approved subunit vaccines now in use (like: NOVAVAX).

Response:

Previously published works on the use of RBD as a vaccine against SARS-CoV-2 have performed immunization using aluminum-based adjuvants and evaluating the protein for different purposes such as, immunogenic comparison with different domains of the spike protein (doi: 10.1080/22221751.2020.1821583), synthetic optimization of the RBD domain to improve its properties (https://doi.org/10.1101/2021.03.03.433558) or as a booster to improve immunity against other variants of the virus (https://doi.org/10.1186/s12985-021-01737-3). However, in our work the focus is to evaluate the immunogenicity of RBD with two oil-based adjuvants, demonstrating that squalene stimulates earlier antibody generation compared to an oil-free formulation, as mentioned in lines 31 - 33 of the modified manuscript.

“In the present report, the immunogenicity of the Spike RBD of SARS-CoV-2 formulated with an oil-in-water emulsion and a water-in-oil emulsion with squalene was evaluated in mice and hamsters.”

Similarly, the introduction to the manuscript on lines 78 - 82 mentions the reported disadvantages of using aluminum adjuvants.

“Alum-based adjuvants are not highly effective in stimulating the cellular immune response of either Th1 or Th2 [16]. These adjuvants require improvements in their concentration and the type of aluminum used to generate a cellular-type immune response; however, these could cause necrosis or tissue damage in the inoculation area [17].”

As well as in the discussion section on lines 496 - 498, this aspect is again emphasized.

“In the present study, the immunization of the RBD sub-domain mixed with two different oil-based adjuvants demonstrate that squalene improves immunogenicity by eliciting an earlier humoral response in mice and hamster.”

We understand that this approach was not clear in the first manuscript submitted, for this reason the title of the study has been modified to: "Squalene in oil-based adjuvant improves the immunogenicity of SARS-CoV-2 RBD and confirms safety in animal models". For the same purpose, the representation of Figures 5 and 7 has been modified, giving it a greater focus on the comparison of the immunogenicity of RBD mixed with the two adjuvants evaluated.

Figure 5. Detection of specific antibodies against RBD in mice. Immunized mice were bled at 0, 15, 30 and 45 days post immunization. All sera were obtained by low-speed centrifugation. Serum samples were processed to detect specific antibodies against SARS-CoV-2 RBD protein using indirect ELISA assay. (A) Group immunized with 20 µg of RBD mixed with A1 and A3 (B) Group immunized with RBD 50 µg of RBD mixed with A1 and A4. Two-way ANOVA and post-hoc Tukey’s test were performed. *: P<0.01

Figure 7. Evaluation of cellular immunity in mice vaccinated with purified RBD. Mice were immunized with 20 and 50 µg of RBD using two different adjuvants (A1 and A3) at 0, 15 and 30 days post immunization. On day 45 post-immunization mice were euthanized and spleens were processed. (A) Percentage of CD4 and CD8 positive cells by flow cytometry, between the groups immunized (n=3, except the adjuvant control). (B) IFN-γ ELISPOT of splenocytes between the groups immunized (n=3, except the adjuvant control). (C) Intracellular staining of Th1 cytokines (IFN-γ, TNF-α and IL-2) of splenocytes stimulated with RBD (n=3, except the adjuvant control). ns: not significant (P>0.05).

The modified Figures 5 and 7 have been re-submitted following the editorial guide of the journal. In addition, the description of these images has been modified in the results section on lines 422 - 431:

“In order to compare the capacity of both oil adjuvants to enhance the immune response to RBD, two amounts of RBD were administered with each adjuvant in mice. Specific antibodies were detected in both groups immunized with each adjuvant. At 15 days post immunization, antibody levels of the group immunized with adjuvant 3 were higher than the group with adjuvant 1, either with the 20 µg and 50 µg dose. However, after the first booster the levels of antibodies generated with both adjuvants were similar in the two doses of RBD evaluated (Fig 5), indicating that the early generation of antibodies in A3 concerning A1 was independent of the dose of protein administered. Control groups immunized with each adjuvant and PBS had baseline reactivities throughout the evaluation time.”

As well as in lines 457 – 469

“The cellular immunity stimulated with the purified RBD in mice was evaluated on day 45 after the first immunization. For adjuvant A1, the percentage of CD4+ and CD8+ T cells increased proportionally to the dose of RBD administered. However, in A3 group the percentage of cells decreased when the dose of RBD was increased (Fig 7A). Regarding the production of Th1-type cytokines (Fig 7C), the increase in CD8+ T cells secreting IFN-γ, TNF-α and IL-2 for A1 and A3 groups decreased according to the administered dose of RBD. However, CD4+ T cells producing these Th1-type cytokines were not detected in all groups. Regarding the secretion of IFN-γ in splenocytes stimulated with purified RBD using the ELISPOT technique (Fig 7B), the adjuvant A3 stimulated a greater number of cells directly proportional to the administered dose. Although there was a difference in the mean values of each group, when the statistical analysis was performed there was not a significant difference between them in all the evaluations performed.”

The figures 3, 4 and 6 have been modified in order to improve the disposition of the lane numbers and the molecular weight markers. The data of these figures has not been modified.

2. Why Authors didn’t test the recombinant purified RBD protein antigenicity against the convalescent COVID-19 patient?

Response:

Unfortunately, despite all the efforts made, ethical clearance for the use of convalescent human sera has not been obtained to date. This limited the use of these biological samples for the purposes of the study.

3. What is the source of RBD that used as a positive control in bounding with Vero-E6 cell?

Response:

We have included the origin of the RBD used as a positive control, the modified paragraph can be found on lines 397 - 399 of the corrected manuscript.

“On the other hand, commercially available recombinant RBD expressed in insect cells (Cat No. Z03479, GenScript) was used as a positive control.”

4. Some examples for sentences need careful editorial review: Some sentences in “Abstract” are too long and containing repetition: from line 38-42 (purified RBD mentioned two times in the same sentence and also the meaning of stimulation), please rewrite in a shorter way.

Response:

The Abstract has been modified according to the reviewer's comments and to give more emphasis to the comparison of the immunogenicity of RBD with two different oil adjuvants.

5. Please unify the use the abbreviation of “W/O” with or without slash; hours (completed or abbreviated); minutes; temperature; room temperature

Response:

All abbreviations have been unified as per the reviewer’s request.

6. Major comments:

a. Please make “space” in line 51 “atypical”, line 55 “add for after urgent”. Respuesta:

b. “°C” with or without space; missing some ending dots like in line 158 after membrane; at the end of line 503.

c. Remove the “dot” after then in line 309

d. Eliminate the word “aseptically” in line 265

e. The word “while” is misleading in line 499

Response:

We thank the reviewer for all these remarks, all of these have already been remedied in the new version of the corrected manuscript.

---

## [Decision Letter · Decision Letter 1]

20 May 2022

PONE-D-21-36362R1Squalene in oil-based adjuvant improves the immunogenicity of SARS-CoV-2 RBD and confirms safety in animal modelsPLOS ONE

Dear Dr. Zimic,

Thank you for submitting your manuscript to PLOS ONE. After careful consideration, we feel that it has merit but does not fully meet PLOS ONE’s publication criteria as it currently stands. Therefore, we invite you to submit a revised version of the manuscript that addresses the points raised during the review process.

We look forward to receiving your revised manuscript.

Kind regards,

Paulo Lee Ho, Ph.D.

Academic Editor

PLOS ONE

Journal Requirements:

Reviewers' comments:

Reviewer's Responses to Questions

**Comments to the Author**

1. If the authors have adequately addressed your comments raised in a previous round of review and you feel that this manuscript is now acceptable for publication, you may indicate that here to bypass the “Comments to the Author” section, enter your conflict of interest statement in the “Confidential to Editor” section, and submit your "Accept" recommendation.

Reviewer #2: All comments have been addressed

Reviewer #3: All comments have been addressed

2. Is the manuscript technically sound, and do the data support the conclusions?

Reviewer #2: Yes

Reviewer #3: Yes

3. Has the statistical analysis been performed appropriately and rigorously? 

Reviewer #2: Yes

Reviewer #3: Yes

4. Have the authors made all data underlying the findings in their manuscript fully available?

Reviewer #2: Yes

Reviewer #3: Yes

5. Is the manuscript presented in an intelligible fashion and written in standard English?

Reviewer #2: Yes

Reviewer #3: Yes

6. Review Comments to the Author

Reviewer #2: I have reviewed the manuscript "Squalene in oil-based adjuvant improves the immunogenicity of SARS-CoV-2 RBD and confirms safety in animal models" which i previously reviewed in its first version and I agree to be published in its current version. The authors have responded to all comments convincingly and have also made all the required modifications. My best regards

Reviewer #3: This is an interesting article on the use of RBD produced in insect cells and formulated in emulsions with squalene as a potential vaccine against covid-19.

The questions raised by the reviewers seem to me to have been adequately answered by the authors. However, there are two additional small issues that I would like to point out: line 121 mentions the use of a tail with 10 histidine residues in the construction of the RBD, however, in figure 3, the scheme indicates 6 residues. Please correct.

Also, please review the symbol for kilo (lowercase k) throughout the text.

7. PLOS authors have the option to publish the peer review history of their article (what does this mean?). If published, this will include your full peer review and any attached files.

Reviewer #2: **Yes: **Reda Salem

Reviewer #3: No

---

## [Author Response · Author response to Decision Letter 1]

23 May 2022

-Reviewer #2: I have reviewed the manuscript "Squalene in oil-based adjuvant improves the immunogenicity of SARS-CoV-2 RBD and confirms safety in animal models" which i previously reviewed in its first version and I agree to be published in its current version. The authors have responded to all comments convincingly and have also made all the required modifications. My best regards

 Response:

 We thank Dr. Reda Salem for the comments that made possible to improve our manuscript.

-Reviewer #3: This is an interesting article on the use of RBD produced in insect cells and formulated in emulsions with squalene as a potential vaccine against covid-19.  The questions raised by the reviewers seem to me to have been adequately answered by the authors. However, there are two additional small issues that I would like to point out: line 121 mentions the use of a tail with 10 histidine residues in the construction of the RBD, however, in figure 3, the scheme indicates 6 residues. Please correct. Also, please review the symbol for kilo (lowercase k) throughout the text.

 Response:

 We are grateful with the reviewer for detecting these typos. All of them have been corrected (line 121) throughout the text.

---

## [Editor Report · Decision Letter 2]

31 May 2022

Squalene in oil-based adjuvant improves the immunogenicity of SARS-CoV-2 RBD and confirms safety in animal models

PONE-D-21-36362R2

Dear Dr. Zimic,

We’re pleased to inform you that your manuscript has been judged scientifically suitable for publication and will be formally accepted for publication once it meets all outstanding technical requirements.

Kind regards,

Paulo Lee Ho, Ph.D.

Academic Editor

PLOS ONE
---

## [Editor Report · Acceptance letter]

12 Aug 2022

PONE-D-21-36362R2 

Squalene in oil-based adjuvant improves the immunogenicity of SARS-CoV-2 RBD and confirms safety in animal models 

Dear Dr. Zimic:

I'm pleased to inform you that your manuscript has been deemed suitable for publication in PLOS ONE. Congratulations! Your manuscript is now with our production department. 

Kind regards, 

on behalf of

Dr. Paulo Lee Ho 

Academic Editor

PLOS ONE